# Valproic Acid Impacts the Growth of Growth Plate Chondrocytes

**DOI:** 10.3390/ijerph17103675

**Published:** 2020-05-22

**Authors:** Hueng-Chuen Fan, Shih-Yu Wang, Yi-Jen Peng, Herng-Sheng Lee

**Affiliations:** 1Department of Pediatrics, Tungs’ Taichung Metroharbor Hospital, Taichung 435, Taiwan; t11578@ms.sltung.com.tw; 2Department of Medical Research, Tungs’ Taichung Metroharbor Hospital, Taichung 435, Taiwan; 3Jen-Teh Junior College of Medicine, Nursing and Management, Miaoli 35053, Taiwan; 4Department of Life Sciences, National Chung Hsing University, Taichung 402, Taiwan; 5Department of Pathology, Tri-Service General Hospital, National Defense Medical Center, Taipei 11490, Taiwan; shihyu0423@gmail.com (S.-Y.W.); yijen0426@gmail.com (Y.-J.P.); 6Department of Pathology and Laboratory Medicine, Kaohsiung Veterans General Hospital, Kaohsiung 81362, Taiwan

**Keywords:** growth plate chondrocyte, valproic acid, apoptosis, cyclooxygenase 2, histone acetylation

## Abstract

A range of bone abnormalities including short stature have been reported to be associated with the use of antiepileptic drugs (AEDs) in children. Exactly how AEDs impact skeletal growth, however, is not clear. In the present study, rat growth plate chondrocytes were cultured to study the effects of AEDs, including valproic acid (VPA), oxcarbazepine (OXA), levetiracetam (LEV), lamotrigine (LTG), and topiramate (TPM) on the skeletal growth. VPA markedly reduced the number of chondrocytes by apoptosiswhile other AEDs had no effect. The apoptosis associated noncleaved and cleaved caspase 3, and caspases were increased by exposure to VPA, which up-regulated cyclooxygenase 2 (COX-2) mRNA and protein levels likely through histone acetylation. The COX-2 inhibitor NS-398 attenuated the effects of VPA up-regulating COX-2 expression and decreased VPA-induced caspase 3 expression. The use of VPA in children should be closely monitored or replaced, where appropriate, by AEDs which do not apparently affect the growth plate chondrocytes.

## 1. Introduction

Epilepsy is a chronic condition characterized by the recurrence of seizures not provoked by a metabolic or toxic disease or fever [1]. The estimated prevalence rates of epilepsy in the US, Europe, Asia, and Taiwan are 6.8 per 1000, 5.5 per 1000, 1.5 to 14 per 1000 [2], and 2.8 per 1000 [3], respectively, and the condition affects almost 50 million people worldwide [4]. Although seizures can spontaneously remit in some cases, lifelong antiepileptic drug (AED) treatment is always necessary for those with refractory epilepsy [5]. However, this creates a medical dilemma as prolonged AED administration is associated with a number of undesirable side effects including endocrine and metabolic disorders, psychiatric and behavioral disorders, and drug interaction side effects [6,7,8]. A range of bone abnormalities such as an increased risk of fractures, abnormal dentition, rickets, osteomalacia, and short stature have also been reported to be linked to the use of AEDs in children and may have seriouseffects on the quality of life of those affected by them [9,10,11].

The diagnosis and treatment of epilepsy most commonly occur in childhood when bone growth is at its maximum. The processes involved in linear growth are very complicated and are regulated by several factors, such as endocrine mechanisms, paracrine mechanisms, proinflammatory cytokines, cartilage extracellular matrix regulation pathways, etc. [12]. The unwanted effects of AEDs on bone health are likely to be related to linear growth. Findings such as low serum calcium and 25-hydroxyvitamin D (25[OH]D) levels as well as decreased bone mineral density (BMD) have been frequently reported in epileptic children treated with AEDs [10,13,14,15,16,17].

The mechanisms by which AEDs influence skeletal metabolism are not clear. Reports suggest that AEDs act as cytochrome P450 enzyme inducers to enhance the catabolism of vitamin D to inactive metabolites, reducing calcium levels [18,19,20]. However, several studies have failed to identify a link between the use of AEDs and altered vitamin D levels [21,22], hypocalcemia [23,24,25], or changes in BMD [26]. Valproic acid (VPA), an effective and commonly used AED, is generally thought to be an inhibitor rather than an inducer of drug-metabolizing enzymes [27], and so the recognized adverse effects of this AED on the skeletons of epileptic patients [11,28,29,30,31,32,33,34] probably involve other mechanisms. Our previous work has demonstrated that VPA treatment is associated with the impairment of statural growth in pediatric patients with epilepsy [35]. Moreover, VPA has been shown to inhibit longitudinal bone growth by reducing cartilage formation and stimulating ossification of the growth plate [36]. In line with this finding, our initial in vitro studies indicated that VPA appeared to reduce the proliferation of rat growth plate chondrocytes, indicating a possible cause of the reduction in longitudinal growth in vivo [35]. 

The current study was conducted to investigate in more detail the impact of AEDs on growth plate chondrocytes and to explore whether recently recognized effects of VPA on inflammatory pathways [37,38,39,40,41] and histone acetylation [42] underlie the identified effects on chondrocytes.

## 2. Materials and Methods

### 2.1. Ethics Statement 

All animal experiment protocols and surgical procedures were approved by the local institutional animal care and use committee, Tri-Service General Hospital, and the National Defense Medical Center, Taipei, Taiwan, ROC (IACUC-12-233). All surgeries were conducted under isoflurane anesthesia. Buprenorphine (0.05 mg/kg injected subcutaneously) was postsurgically provided as an analgesic. All experiments undertaken in this work complied with relevant guidelines and regulations. The mice were housed in an animal facility with a 12-h day/night cycle and adequate temperature and humidity controls. Astandard rodent chow diet [Na^+^: 0.4% (w/w); K^+^: 1.0% (w/w); Ca^2+^: 0.9% (w/w)] and plain drinking water ad libitum were provided for 12 to 14 weeks. 

### 2.2. Reagents and Antibodies

Dulbecco’s modified Eagle’s medium/Nutrient Mixture F-12 HAM medium (DMEM/F-12) was obtained from GIBCO (Life Technologies, Carlsbad, CA, USA). Dimethylsulfoxide (DMSO), fetal bovine serum (FBS), and tetrazolium methylthiotetrazole (MTT) were obtained from Sigma (St. Louis, MO, USA). All other reagents were tissue culture grade and obtained from Sigma. Depakine (valproic acid—VPA) solution was purchased from Sanofi Winthrop Industrie, France; Trileptal (oxcarbazepine—OXA) suspension was purchased from Novartis, France; Keppra (levetiracetam—LEV) solution was purchased from GlaxoSmithKline, Holland; Lamictal (lamotrigine—LTG; 50 mg tablet) was purchased from GlaxoSmithKline, Holland; and Topamax (topiramate—TPM; 100 mg tablet) was purchased from Janssen-Cilag, Switzerland. Based upon therapeutic plasma concentrations of the respective drug in patients, the recommended concentrations of those drugs were used: VPA, 415 μM (60 μg/mL); OXA, 30 μM (7 μg/mL); LEV, 220 μM (37 μg/mL); LTG, 20 μM (5 μg/mL); and TPM, 30 μM (10 μg/mL). The antibodies used were obtained from the following sources: anti-tubulin-α Ab-2 monoclonal antibody (NeoMarkers, Fremont, CA, USA), anti-cyclooxygenase-2 polyclonal antibody (NeoMarkers), anti-caspase-3 polyclonal antibody (Cell Signaling Technologies, Danvers, MA, USA), anti-cleaved-caspase-3 polyclonal antibody (Cell Signaling Technologies), anti-histone-H3 polyclonal antibody (Upstate Biotechnology, Lake Placid, NY, USA), anti-histone-H4 polyclonal antibody (Cell Signaling Technologies), anti-mouse-HRP polyclonal antibody (Dakocytomation, Glostrup, Denmark), anti-rabbit-HRP polyclonal antibody (Dakocytomation), and horseradish peroxidase-conjugated antimouse IgG (Sigma).

### 2.3. Cell Culture

Male 3-week-old Sprague–Dawley rats (50–60 g each) were purchased from BioLASCO Taiwan (Taipei, Taiwan). The epiphyseal growth plate of the tibia was aseptically collected by cleaning the cartilage plate of muscular tissue, periosteum, and perichondrium. The proximal epiphysis was dissected with a sharp scalpel, and the cartilage plate was detached distally from the tibial metaphysis. Chondrocytes, which were obtained from tibial growth plates digested in 3 mg/mL of collagenase type H (Sigma)for 3 h at 37 °C, were cultured as monolayers in DMEM/F-12 medium, 10% heat-inactivated fetal bovine serum (GIBCO), 100 IU/mL penicillin (GIBCO), and 100 mg/mL streptomycin (GIBCO) in 5% CO_2_ and 95% air at 37 °C. Once fully confluent, the cells were harvested using trypsin-EDTA (GIBCO) and subcultured at a 1:3 ratio. The chondrocytes were routinely checked for positive anti-S100 protein (data not shown). A mycoplasma ELISA kit (Roche, Mannheim, Germany) was routinely used for thedetectionof contamination bymycoplasma in cell cultures. 

### 2.4. AED Treatment and MTT Assay 

For the AED treatment protocol, growth plate chondrocytes grown at passages 2 or 3 were dispensed at 1 × 10^4^ cells/mL in 96-well plates. The cell culture medium was replaced daily by one containing fresh complete medium, fresh complete medium with 0.1% DMSO as a vehicle, or fresh complete medium with the recommended concentration of the respectiveAED. For MTT assays, cells in each well were incubated with 200 μL of MTT (0.5 mg/mL) for 3 h at 37 °C. The medium was discarded and the resultant formazan crystals retained in the cells were dissolved in DMSO (200 μL). After shaking for 5 min, the absorbance of the cells in each well wasdetectedat 540 nm with a microplate reader (μQuant, BIO-TEK Instruments Inc., Winooski, VT, USA) installed withthe KC Junior analysis software, version 1.5 (BIO-TEK Instruments). The percentage of viable cells (%) was defined as [(A − B)/(C − B)] × 100 where A = OD560 of the treated sample, B = OD560 of the background absorbance, and C = OD560 of the reference cells not exposed to the chemical compound being tested. Background absorbance was measured by lysing the reference cells with 1% DMSO before applying MTT. To investigate the dose-effect of VPA, cells were treated daily with 0, 30 μg/mL (0.5X), 60 μg/mL (1X), 150 μg/mL (2.5X), 300 μg/mL (5X), 450 μg/mL (7.5X), and 6000 μg/mL (10X) VPA for 5 days.

### 2.5. RNA Extraction and Real-Time Polymerase Chain Reaction (RT-PCR)

Total RNA of the cultured growth plate chondrocytes was isolated and purified using TRIzol^®^ RNA Isolation Reagents (Invitrogen, Liverpool, NY, USA). For the synthesis of first-strand cDNA, 2 μg of total RNA was collected for a single-round reverse transcription reaction performed using a High Capacity cDNA Reverse Transcription Kit (Applied Biosystems, Foster City, CA, USA). cDNAs were exponentially doubled under the conditions of 95 °C for 10 min, 40 cycles of 95 °C for 15 s, and 60 °C for 60 s using the Power SYBR^®^ green PCR master mix (Applied Biosystems) and a StepOne™ Real-Time PCR System (Applied Biosystems). Simultaneousamplification of β-actinwas used as an internal control to normalize the various mRNA levels in the samples to quantify the changes of gene expression using the 2−ΔΔCt formula. The specific primers used in this study are shown in Table 1.

### 2.6. Protein Extraction and Western Blotting

After VPA treatment, cells were immediately immersedin ice-cold PBS and lysed in situ for 15 min with ice-cold radio-immunoprecipitation assay (RIPA) lysis buffer (Thermo Pierce, Thermo Fisher Scientific Inc., Rockford, IL, USA) containing 100 μM Na_3_VO_4_ and a protease inhibitor cocktail tablet (Roche Diagnostics, Mannheim, Germany). After centrifugation at 13,000 rpm for 15 min, the cell lysates were collected. Equal amounts of protein from these lysates were loaded onto 10% SDS-polyacrylamide gel to separate various proteins contained in the lysates. The separated proteins were then transferredto polyvinylidene fluoride (PVDF) membranes (Merck Millipore, Darmstadt, Germany). The membranes were blocked with 2% bovine serum albumin (BSA) in Tris buffered saline buffer with Tween 20 (TBST) (12.5 mM Tris/HCl, pH 7.6, 137 mM NaCl, 0.1% Tween 20) at 4°C overnight. After being washed three times with TBST, the blots were stained with primary antibody, which was diluted in TBST. Following further washing, the blots were stained with HRP-labeled secondary antibody before undergoing detection using an enhanced chemiluminescence Western blotting detection system (Merck Millipore) according to the manufacturer’s instructions. The membranes were scanned and analyzed by densitometry (VisionWorks LS, UVP, LLC, Upland, CA, USA). Enhanced chemiluminescence (ECL) kit was supplied by Amersham Pharmacia Biotech (Uppsala, Sweden). All other materials were purchased from Sigma unless otherwise stated. 

### 2.7. Flow Cytometry

Apoptosis was measured using the fluorescein isothiocyanate-labeled AnnexinV/7-amino-actinomycin D (7-AAD) Apoptosis Detection kit (BD, Pharmingen^TM^, San Jose, CA, USA) according to the manufacturer’s instructions. 1 × 10^6^ rat chondrocytes, which were stimulated with 60 μg/mL VPA for 5 days, were labelled with 5 μL of 7-AAD and 5 μL of Annexin-V for 15 min at 20–22 °C. The cells were then transferred to BD tubes and centrifuged at 1,000 rpm for 10 min at 4 °C. Data acquisitions were conducted using flow cytometry. All cytometry analyses in this study were performed on a FACSCalibur and using CELLQUEST software (version 6.0, BD Bioscience Pharmingen Inc., San Diego, CA, USA).

### 2.8. Statistical Analysis

All values were expressed as mean ± standard deviation (SD). The quantification data of the mRNA and protein expression levels was analyzed by Student’s t tests. Comparisons of the data from cell proliferation studies were analyzed by ANOVA. A *p* value <0.05 was considered statistically significant.

## 3. Results

### 3.1. VPA Markedly Reduces the Number of Chondrocytes

Rat growth plate chondrocytes were treated daily with 60 μg/mL VPA, 7 μg/mL OXA, 37 μg/mL LEV, 5 μg/mL LTG, or 10 μg/mL TPM for 5 days, and the number of cells was then assessed by MTT assay. There was a significant reduction in the number of chondrocytes following 5 days of treatment with VPA, but not with the other AEDs, in comparison to the untreated control (control 100% vs. VPA 72.65 ± 6.68%, *p* = 0.0064) (Figure 1). A dose-response experiment conducted over a concentration range of 0–600 μg/mL for 5 days of consecutive daily VPA treatments showed a significant decrease in chondrocyte number over the dose range, with a statistically significant effect being demonstrated at the lowest dose used (30 μg/mL) (Figure 2).

### 3.2. VPA Has No Effect on Cartilage Matrix Gene Expression

The chondrocyte cartilage matrix genes including collagen type IIa1 (Col2a1), type Xa1(Col10a1), and aggrecan (ACAN) genes were analyzed following 5 days with or without VPA (60 μg/mL). The results showed no changes in these three genes (Figure 3).

### 3.3. VPA Induces Chondrocyte Apoptosis, Noncleaved and Cleaved Caspase 3 Expression

Rat growth plate chondrocytes were treated with VPA for 5 days, followed by labeling with annexin-V/7-AAD. Compared with the control, untreated group, the number of early apoptotic cells in the VPA-treated group was significantly higher (control 3.61 ± 1.09% vs. VPA 14.35 ± 2.62%, *p* < 0.001) (Figure 4A,B). Furthermore, Western blot analysis showed that 5 days of VPA treatment prominently increased the levels of caspase 3 (1.39 ± 0.07 fold, *p* < 0.001) (Figure 4C,D) and cleaved caspase 3 (1.46 ± 0.29 fold, *p* = 0.021) (Figure 4E,F).

### 3.4. VPA Increases COX-2 Expression in Growth Plate Chondrocytes

As COX-2 is a major proinflammatory signaling molecule that is involved in regulating chondrocyte apoptosis, we investigated whether VPA treatment regulated the COX-2 expression of rat growth plate chondrocytes in our system. Chondrocytes were treated daily with VPA and other AEDs for 5 days. Gene expression analysis showed that VPA, but not OXA, LEV, LTG, and TPM, significantly up-regulated the expression of both COX-2 mRNA (1.61 ± 0.05 fold, *p* = 0.0137) (Figure 5A,B) and protein (2.38 ± 0.49 fold, *p* = 0.0478) (Figure 5C,D). To optimize the VPA effects on the growth plate chondrocytes, we discovered that the 5-day time course of the VPA treatment induced the COX-2 mRNA levels in the 4days (4.64 ± 0.25 fold,), and the 5-days group (4.29 ± 0.235 fold) showed significantly higher than the first day group (1.61 ± 0.18 fold) (Figure 6).

### 3.5. VPA Increases Histone Acetylation in Growth Plate Chondrocytes 

To examine whether VPA induced histone acetylation in growth plate chondrocytes, cells were harvested after treatment with VPA for 4 days. The levels of total histone H3 and H4 protein were unaffected by the VPA treatment. In contrast, the VPA treatment caused a prominent increase in the levels of acetylated histone H3 (4.37 ± 0.66 fold, *p* < 0.05) and H4 (2.31 ± 0.16 fold, *p* < 0.001) in comparison to untreated controls (Figure 7).

### 3.6. Apoptosis Induced by VPA on Rat Growth Plate Chondrocyte Are Through COX-2 Dependent Inflammation Pathway 

To investigate the role of COX-2 in regulating the effects of VPA on chondrocyte growth in culture, rat growth plate chondrocytes cells were concurrently treated with NS-398, a specific COX-2 inhibitor, and VPA for 4 days. The expression of COX-2 at both the gene and protein level was significantly suppressed when the chondrocytes were coincubated with VPA and NS-398 (Figure 8A,B). Over the time course of the experiment, the cotreatment of NS-398 with VPA caused anotable increase in the number of viable chondrocytes in culture from 73.62 ± 4.36% with VPA treatment alone to 87.57 ± 3.63% under cotreatment of NS-398 with VPA (*p* < 0.05, compared with VPA-alone treatment group) (Figure 8C). Similarly the expression of cleaved caspase 3, which was increased by 1.71 ± 0.23 fold in the VPA-alone treatment group (*p* < 0.005, compared with control), was significantly lower (1.24 ± 0.08 fold, *p* < 0.05) in the NS-398/VPA cotreatment group (Figure 8D).

## 4. Discussion

In the current study, we investigated how 5 consecutive days of treatment with one of five AEDs, namely, VPA, LEV, OXA, LTG, or TPM, affected the proliferation of cultured growth plate chondrocytesin vitro, and showed that VPA, unlike the new-generation AEDs, significantly reduced chondrocyte proliferation. These findings are consistent withour own clinical observations [35] and several previous reports [11,22,28,31,36,43,44,45,46]. More specifically, we discovered that 5 days of treatment with 60 μg/mL of VPA, which is within the normal therapeutic range of VPA for epilepsy (50–100 μg/mL) [47], significantly inhibited chondrocyte proliferation. By examining the chondrocyte viability percentage under treatment with various concentrations of VPA, we further found that the inhibitory effects of VPA showed a dose-dependent trend (for doses from 0 to 150 μg/mL) toward decreased chondrocyte growth velocity, with even the relatively low dose of 30 μg/mL of VPA showing a significant inhibitory effect on chondrocyte replication.In comparison with one model by Wu et al. [36] in more detail which carried out the effects of VPA in prenatal stage with reduction of chondrocyte hypertrophy, our study was based on clinical data presented in our previous work [35], which showed that VPA could reduce growth in children. The current study goes further to examine the effects of VPA in childhood with reduction of chondrocyte proliferation. The hypertrophic mechanisms involved by VPA (in different stage of anatomic site) need to be further elucidated.

In basic terms, the mechanisms of bone formation involveboth endochondral and intramembranous ossification. The former contributes to the growth of the axial and appendicular bones, while the latter is related to the growth of the craniofacial bones [48]. Endochondralossification, which continues throughout the period of growth, entailssequential biological processes, in which chondrocytes proliferate and undergo hypertrophy to generate new cartilage at the growth plate of a long bone [49]. Simultaneously, the growth plate is invaded from the bony metaphysis by blood vessels and bone cell precursors, which remodel the cartilage into bone. Such remodeling appears to be triggered by apoptosis or cell death of the hypertrophic chondrocytes adjacent to the metaphyseal bone. The coordinated processes of chondrogenesis and ossification lead to long bone growth [50].

Chondrogenesis is one of the earliest and most important morphogenetic steps in the skeletogenesis of vertebrates [51]. ACAN, Col 2, and Col 10 are the cartilage matrix genes [51,52]. A previous study showed that ACAN mRNA levels are significantly reducedafter exposure to VPA [53], while another study found that cultured human chondrocytes produce less Col 2 mRNA and decrease the synthesis of Col 2 collagen after exposure to VPA [54]. Meanwhile, Coghlan et al. discovered that the concentration of Col 10 is correlated with skeletal growth velocity [55]. However, our data in the present study indicated that VPA had no effect on Col 2a1, Col 10a1, or ACAN expression in growth plate chondrocytes, suggesting that other mechanisms may be responsible.

Inflammation is associated with bone growth [56]. COX-2 is a pivotal proinflammatory mediator in articular cartilage [57] but has also been shown to be linked to chondrocyte hypertrophic differentiation in rabbit growth plates, whereas COX-2, along with iNOS, is involved in injury-induced inflammatory responses and may facilitate mesenchymal stromal cell differentiation to chondrocytes during bony repair of growth plate injury sites [58].As such, the increased expression of COX-2 subsequent to VPA treatment could be anticipated to have a positive effect on growth plate chondrocytes, rather than the negative effect that we have identified. The growth plate is, however, a complex tissue with several specialized chondrocyte zones which interact through a variety of positive and negative feedback loops [52], with coordination between chondrocyte proliferation, hypertrophy, and cell death, allowing the cartilage to maintain its cellularity and provide the appropriate local environment for osteoblast recruitment and bone growth. As such, while the increased expression of COX-2 in growth plate chondrocytes seen in the present study would be expected to enhance chondrocyte hypertrophy and, thus, chondrocyte proliferation [59], it is possible that that expectation was not met because the monolayer chondrocyte culture model used in the present study does not have the spatial organization or other relevant characteristicsof the growth plate in vivo. Nonetheless, given the association between chondrocyte hypertrophy and caspase 3 activation, which results, in turn, in terminal chondrocyte differentiation and death [60], our observation that 4 consecutive days of VPA treatment significantly increased the expression of caspase-3 and cleaved caspase-3 further suggest the possibility that the adverse effects of VPA on statural growth may occur as a result, at least in part, of enhanced chondrocyte hypertrophy and terminal differentiation that are not balanced by the proliferation and recruitment of chondrocytes. Meanwhile, the newer AEDs investigated in this study (that is, LEV, OXA, LTG, and TPM) that had no significant effects on chondrocyte proliferation also exhibited no up-regulation of COX-2 gene expression. In other words, the results of this study suggest that the effects of VPA on chondrocytes occur through a COX-2-dependent inflammatory pathway that eventually activates caspase 3-dependent apoptosis, leading to a decrease in the number of chondrocyte cells due to an imbalance with the proliferation and recruitment of chondrocytes. The anti-inflammatory effect of NS-398 mitigates this VPA-driven COX-2 overexpression and, in turn, reduces the VPA-induced caspase overexpression.

Furthermore, the results of this study indicated that VPA treatment induces histone acetylation in growth plate chondrocytes, which is likely to have important effects on chondrocyte differentiation and function [61,62]. These effects of VPA on growth plate chondrocyte histone acetylation also indicate likely mechanisms involving terminal differentiation and cell death. VPA has been reported to be an effective histone deacetylase (HDAC) inhibitor at concentrations well within the therapeutic range used for epilepsy [63] and can activate apoptosis in several cell lines. In the current study, we found that VPA caused hyperacetylation of the *N*-terminal tails of histones H3 and H4. HDAC 4, whichacts as a negative regulator of chondrocyte hypertrophy, binds to and inhibits Runx2, an important transcription factor for chondrocyte hypertrophy. Loss of HDAC4 results in the early onset of endochondral bone ossification because ofaccelerated chondrocyte hypertrophy, while the overexpressed HDAC4 arrests hypertrophy and differentiation [64]. Nonetheless, additional studies are required to identify how VPA increased hyperacetylation in our model system and whether it also does so in more physiologically representative models.

This study had several limitations. First, rat growth plate chondrocytes cannot truly reflect in vivo human conditions. Second, the rat growth plate chondrocytes were cultured in monolayer and were not supplemented with ascorbic acid and growth factors, possibly leading to a loss of polygonal morphology and dedifferentiation [65]. A 3D system with such supplementations may be a more physiological setting that betterpreserves thespatial organization of the growth platethan the monolayer culture system used in this study. Third, as in vitro settings cannot simulate complex situations such as the compartmentation and microarchitecture of a joint, using a combination of in vitro and in vivoanimal experiments should be a better strategy for investigating the impacts of VPA onstatural growth. These limitations may have led to some bias in analyzing the effects of VPA on the growth of growth plate chondrocytes in the present study.

## 5. Conclusions

The current study identifies a potential mechanism for the VPA-associated decrease in statural growth in children involving increased COX-2 expression, increased activation of caspase 3, and increased apoptosis in growth plate chondrocytes, with those increases in turn being associated with histone hyperacetylation.As childhood and adolescence are crucial periods for the attainment of peak bone mass, and because the majority of patients with epilepsy are first diagnosed and treated during childhood or adolescence, AEDs, especially VPA, should be carefully used in pediatric and adolescent patients with epilepsy.The use of new generation AEDs that do not appear to have adverse effects on growth plate chondrocyte would be preferable in treating children and adolescents with epilepsy. Although this in vitro study suggests that a variety of pharmacological agents such as COX-2 inhibitors or, potentially, HDAC inhibitors may be used to mitigate the adverse effects of VPA on chondrocyte growth, further research on humans is needed to confirm this experimentally indicated possibility.

## Figures and Tables

**Figure 1 ijerph-17-03675-f001:**
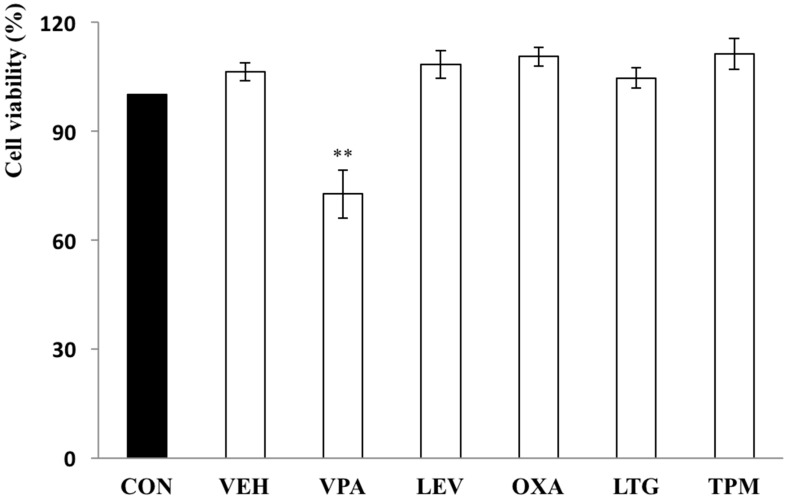
VPA significantly inhibits the proliferation of rat growth plate chondrocytes.Rat growth plate chondrocytes were treated daily with 60 μg/mL VPA, 7 μg/mL OXA, 37 μg/mL LEV, 5 μg/mL LTG, or 10 μg/mL TPM for 5 days. The proliferation of the chondrocytes was assessed by the MTT assay. Among the AEDs tested, rat chondrocytes significantly decreased 72.65 ± 6.68% in the VPA group (*p* = 0.0064) (*n* = 5, ** *p* < 0.01, compared with control). CON—control, culture medium; VEH—vehicle, 0.1% DMSO+culture medium; VPA—valproic acid, 60 μg/mL; LEV—levetiracetam, 37 μg/mL; OXA—oxcarbazepine, 7 μg/mL; LTG—lamotrigine, 5 μg/mL; and TPM—topiramate, 10 μg/mL.

**Figure 2 ijerph-17-03675-f002:**
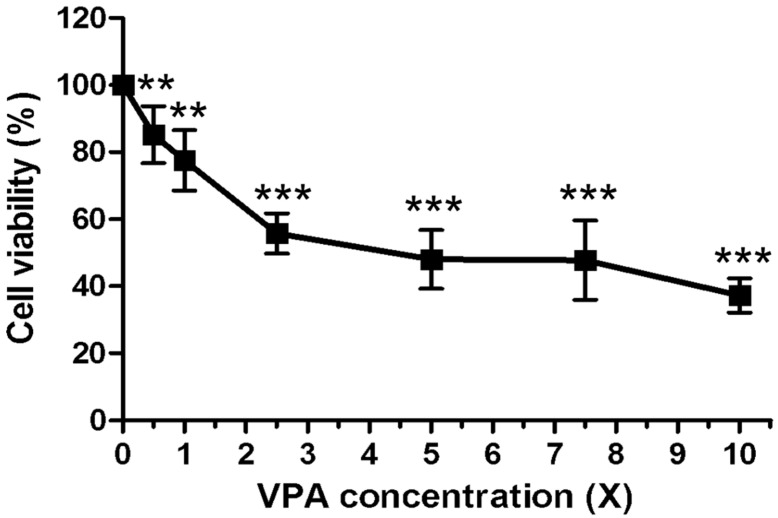
The inhibitory effect of VPA on the growth of rat chondrocytes is dose-dependent. Values are percentages of control. Rat chondrocytes were treated with consecutive 5 days with varying concentrations of VPA (1X = 60 μg/mL) (IC50 ≒ 5X) (*n* = 6, ** *p* < 0.01, *** *p* < 0.001, compared with control).

**Figure 3 ijerph-17-03675-f003:**
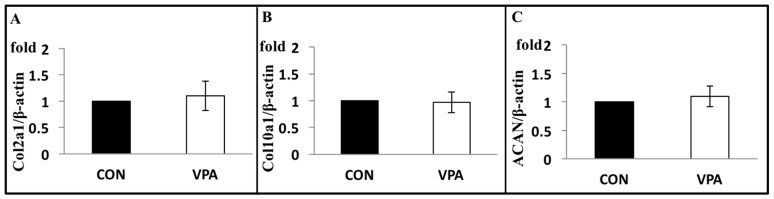
VPA has no effects on the expression levels of cartilage matrix genes of rat growth plate chondrocytes, including (**A**) Col2a1, (**B**) Col10a1, and (**C**) ACAN. Chondrocytes were incubated with or without 60 μg/mL of VPA for 5 days. mRNA expression levels of the β-actin used as an internal control (*n* = 5; *p* > 0.05).

**Figure 4 ijerph-17-03675-f004:**
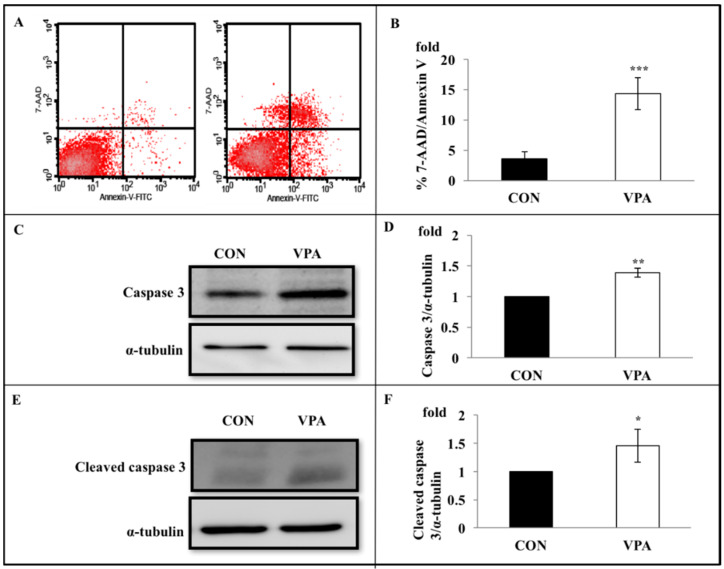
VPA induces rat growth plate chondrocyte apoptosis, caspase 3 expression, and caspase 3 cleavage. Rat growth plate chondrocytes were incubated with 60 μg/mL of VPA for 5 days, followed by labeling with annexin-V andannexinV/7-amino-actinomycin D (7-AAD). (**A**) A representative dot-plot showed annexin-V vs. 7-AAD permeability protocol, with the four major populations of viable and apoptotic cells identified. VPA induced the average percentage of cells in the early stages of apoptosis (annexin-V positive/7-AAD negative) increased from 3.44% to 11.4%. Left: control; right: VPA. (**B**) quantifying four repeated studies showed that VPA significantly induced chondrocytes in the early stages of apoptosis (Control vs. VPA: 3.61 ± 1.09 % vs. 14.35 ± 2.62%; *n* = 4, *** *p* < 0.001). (**C**) The upper blot showed that VPA treatment enhanced the expression levels of caspase 3. The lower blot shows the α-tubulin expression levels that were detected in the same experiment by stripping and reprobing using anti-α-tubulin antibody. (**D**) Collected results from four experiments. Caspase 3 protein band densities (upper blot in **C**) expressed relative to α-tubulin expression levels (lower blot). The ordinate shows caspase 3 protein expression levels normalized to level in absence of VPA. Results showed that VPA increased 1.39 ± 0.07-fold of the caspase 3. α-tubulin is used as an internal control. (*n* = 4, ** *p* < 0.01). (**E**) The upper blot shows that the cleaved caspase 3 expression was increased by VPA treatment. The lower blot shows the α-tubulin expression levels that were detected in the same experiment by stripping and reprobing using anti-α-tubulin antibody. (**F**) Collected results from four experiments. Cleaved caspase 3 protein band densities (upper blot in **E**) expressed relative to α-tubulin expression levels (lower blot). The ordinate shows cleaved caspase 3 protein expression levels normalized to level in absence of VPA. Results showed that VPA increased 1.46 ± 0.29 fold of the cleaved caspase 3. α-tubulin is used as an internal control. (*n* = 4, * *p* < 0.05).

**Figure 5 ijerph-17-03675-f005:**
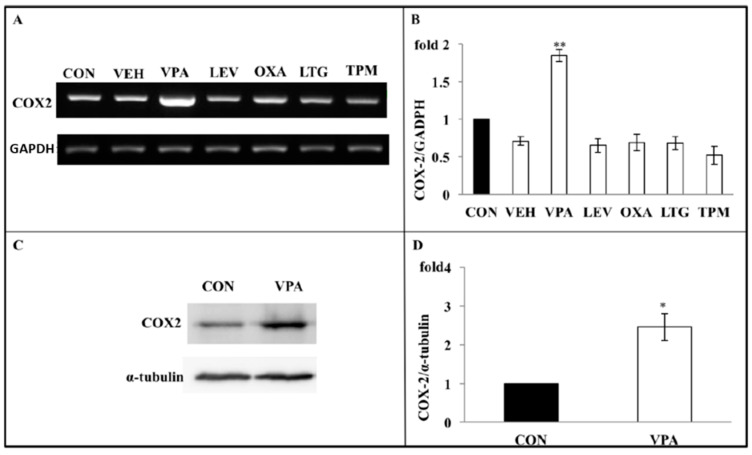
Apoptosis induced by VPA on rat growth plate chondrocyte are throughcyclooxygenase 2 (COX-2) dependent inflammation pathway. (**A**) Following 5 daily AED treatment, including VPA, LEV, OXA, LTG, or TPM, mRNA of the chondrocytes was extracted and amplified with RT-PCR. The upper gel showed that only VPA significantly increased the expression of COX-2 mRNA. GADPH served as internal controls. (*n* = 6, ** *p* < 0.01, compared with control). (**B**) Quantitative data of expression levels of COX-2 mRNA in the experiment A. All bars represent the mean ± S.D. The bars are (from left), CON (control, culture medium); VEH (vehicle, 0.1% DMSO+culture medium); VPA (valproic acid, 60 μg/mL); LEV (levetiracetam, 37 μg/mL); OXA (oxcarbazepine, 7 μg/mL); LTG (lamotrigine, 5 μg/mL); and TPM (topiramate, 10 μg/mL)). (**C**) A representative gel showing that VPA treatment significantly increased COX-2 protein expression. α-tubulin is used as an internal control. (**D**) COX-2 band densities (upper blot in **C**) expressed relative to GADPH expression levels (lower blot). The ordinate shows the normalized COX-2 level in the absence of VPA (*n* = 4, * *p* < 0.05).

**Figure 6 ijerph-17-03675-f006:**
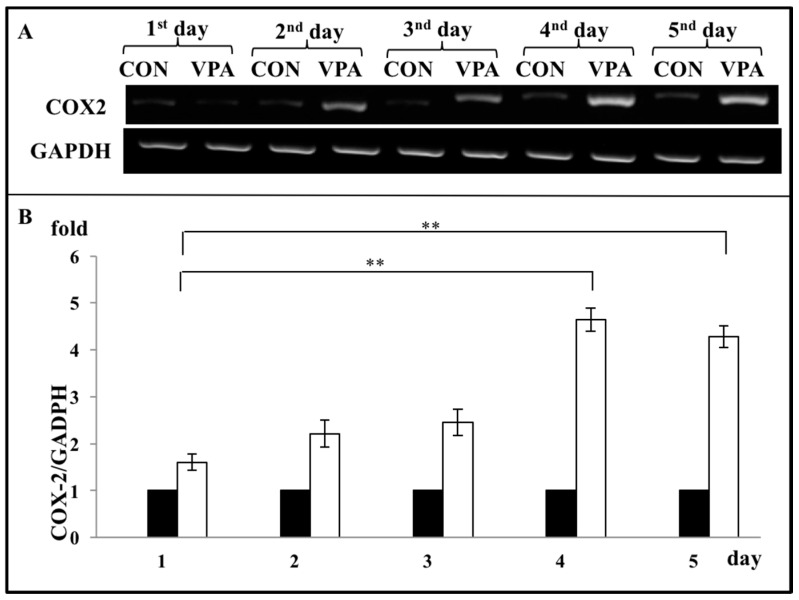
Optimization for VPA treatment on the chondrocytes. Rat growth plate chondrocytes were plated for 5 groups and 60 μg/mL of VPA were daily treated in each from 1 to 5 days. After treatment, each group’s COX-2 mRNA expression levels were analyzed and compared. (**A**) The upper panel of the gel showed COX-2 mRNA levels in the 4 days and 5 days group significantly higher than the first day group. GADPH served as internal controls. (**B**) Quantitative data of expression levels of COX-2 mRNA in the experiment A. All bars represent the mean ± S.D. The black bars are CON (control, culture medium) and the white bars are VPA (valproic acid, 60 μg/mL) (*n* = 7, ** *p* < 0.01).

**Figure 7 ijerph-17-03675-f007:**
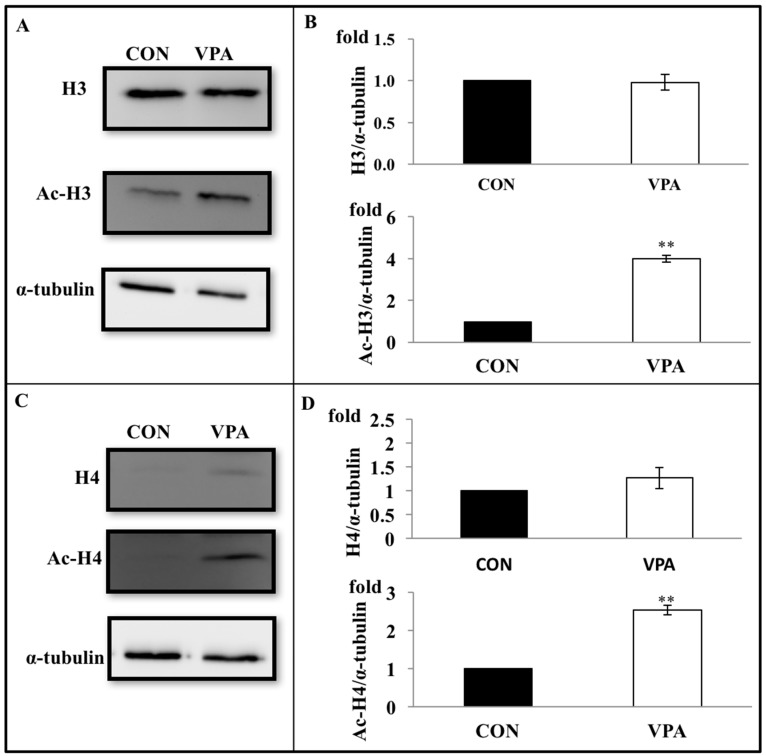
Effects of VPA on histone acetylation. Rat chondrocytes were cultured with daily VPA treatment for 4 days. Protein lysates were isolated and immunoblotted for histone 3 (H3), acetylated histone 3 (Ac-H3), histone 4 (H4), and acetylated histone 4 (Ac-H4). (**A**) A representative gel showing H3, Ac-H3, and α-tubulin expression in rat chondrocytes with and without VPA treatment. α-tubulin wasused as an internal control. (**B**) Proteins of the H3 and Ac-H3 were quantified by densitometry (*n* = 4, ** *p* < 0.01). (**C**) A representative gel showing H4, Ac-H4, and α-tubulin expression in rat chondrocytes with and without VPA treatment. α-tubulin wasused as an internal control. (**D**) Proteins of the H4 and Ac-H4 were quantified by densitometry (*n* = 4, ** *p* < 0.01).

**Figure 8 ijerph-17-03675-f008:**
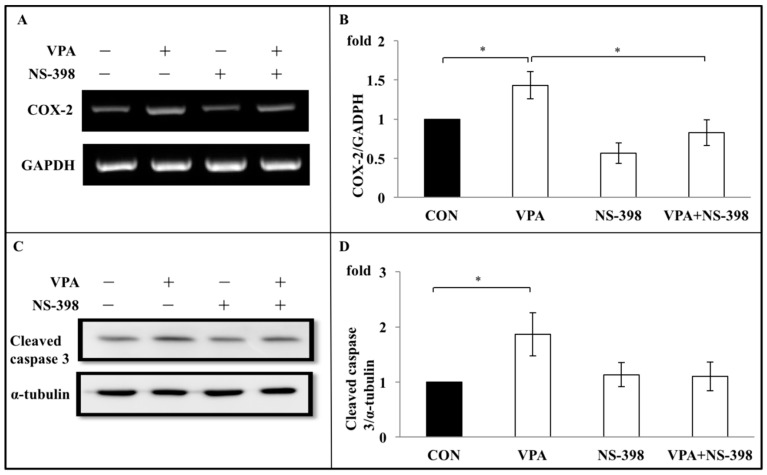
NS-398 abolishes caspase expression increased by VPA. (**A**) A representative gel showed that VPA increasedCOX-2 mRNA expression, while cotreated NS-398 with VPA abolished the increase of COX-2 mRNA expression. NS-398 alone showed no effects on the COX-2 mRNA expression. (**B**) Quantitative data of expression levels of COX-2 mRNA in the experiment in A. All the bars show the mean ± S.E. The bars are (from left), CON (control, culture medium); VPA (valproic acid, 60 μg/mL); NS-398 (50 μM); and VPA + NS-398 (valproic acid 60 μg/mL + NS-398 50 μM). *n* = 4 in each experiment. * *p* < 0.05, (**C**) VPA increased cleaved caspase 3 protein levels, and the increase of the cleaved caspase 3 protein levels was abolished by cotreated NS-398 (50 μM). NS-398 alone showed no effects on the cleaved caspase 3 protein expression. (**D**) Quantitative data of expression levels of cleaved caspase 3 protein levels in the experiment in C. All the bars showed the mean ± S.D. The bars are (from left), CON (control, culture medium); VPA (valproic acid, 60 μg/mL); NS-398 (50 μM); and VPA + NS-398 (valproic acid 60 μg/mL + NS-398 50 μM). *n* = 4 in each experiment. * *p* < 0.05.

**Table 1 ijerph-17-03675-t001:** Oligonucleotide primers for semiquantitative RT-PCR analysis.

mRNA	Sequences	Product Size (Base Pairs)	Accession No.
Rat			
GADPH	5′-GAACGGGAAGCTCACTGGC-3′	70	S67722
	5′-GCATGTCAGATCCACAACGG-3′		
COX-2	5′-CCCTGAAACCTTACACATCGTTT-3′	90	X02231
	5′-TGGCATCGATGTCATGGTAGA-3′		
β-actin	5′-GGAGATTACTGCCCTGGCTCCTA-3′	150	S67722
	5′-GACTCATCGTACTCCTGCTTGCTG-3′		
Col2a1	5′-TCCTAAGGGTGCCAATGGTGA-3′	112	X02231
	5′-GGACCAACTTTGCCTTGAGGAC-3′		
Col10a1	5′-TTCACAAAGAGCGGACAGAGA-3′	143	S67722
	5′-TCAAATGGGATGGGAGCA-3′		
ACAN	5′-TCCGCTGGTCTGATGGACAC-3′	101	X02231
	5′-CCAGATCATCACTACGCAGTCCTC-3′		

GADPH—glyceraldehyde 3-phosphate dehydrogenase; COX-2—cyclooxygenase-2; Col2a1—collagen type IIa1; Col10a1—collagen type Xa1; ACAN—aggrecan. For each primer pair, the top sequence is forward and the bottom sequence is reverse.

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
