# Peer review of "Valproic Acid Impacts the Growth of Growth Plate Chondrocytes"

_ijerph, 2020, doi:10.3390/ijerph17103675_

Round 1

Reviewer 1 Report

The title should be more specific. The title covers a wider range than the results of authors’ research.

Introduction section

Authors should summarize the prior literature on the effects of antiepileptic drugs on longitudinal bone growth and their known mechanisms of action and explain the purpose of this study in detail. For example, valproic acid has been shown to inhibit longitudinal bone growth by reducing cartilage formation and stimulating ossification of the growth plate (J Child Neurol. 2004;19(1):26-30).

In addition, further explanation is needed as to why authors focused on COX-2 expression and histone acetylation

Discussion section

Lines 313–318: Since the subject of this research is how antiepileptic drugs affect skeletal growth in children, the discussion should be focused on the endochondral ossification of the growth plate rather than prenatal development.

Valproic acid is known to reduce chondrocyte hypertrophy (J Child Neurol. 2004;19(1):26-30), which conflicts with authors' suggestion. The prior studies should be comprehensively considered to discuss the differences between the results.

It is necessary to describe the limitation of the monolayer culture model (for example, a loss of polygonal morphology and dedifferentiation).

Conclusion section

The contents of Lines 370–377 and lines 379–387 overlap.

Minor points

Running title (line 4): Use more specific verb than “impact”.

Lines 198–199 and 317–318: Once an abbreviation has been defined, don't repeat the definition again.

Lines 172–174: The sentences should be deleted (editing error).

Lines 167, 249, and 298: The terms “standard error” and “standard deviation” were confused.

Figure 6: Replace 3nd, 4nd, and 5nd with 3rd, 4th, and 5th, respectively.

Author Response

Reviewer A

Comments and Suggestions for Authors

Q: The title should be more specific. The title covers a wider range than the results of authors’ research.

 R: Thank you for your suggestion. We have now revised the title to “Valproic Acid Impacts the Growth of Growth Plate Chondrocytes”

Q: Introduction section

Authors should summarize the prior literature on the effects of antiepileptic drugs on longitudinal bone growth and their known mechanisms of action and explain the purpose of this study in detail. For example, valproic acid has been shown to inhibit longitudinal bone growth by reducing cartilage formation and stimulating ossification of the growth plate (J Child Neurol. 2004;19(1):26-30).

R: Thank you for this insightful comment. As per your suggestion, we have revised the content and added a sentence about VPA has been shown to inhibit longitudinal bone growth by reducing cartilage formation and stimulating ossification of the growth plate [36] in line 63-65.

Q: In addition, further explanation is needed as to why authors focused on COX-2 expression and histone acetylation

 R: Thank you for this insightful comment. We have added the following sentence and citation in line 339 “Inflammation is associated with bone growth [56].” Relatedly, we have also explained the role of COX-2 in inflammation in the following sentences.

We also explained the role of histone acetylation as follows in lines 368-370: “Furthermore, the results of this study indicated that VPA treatment induces histone acetylation in growth plate chondrocytes, which is likely to have important effects on chondrocyte differentiation and function.”

Discussion section

Q: Lines 313–318: Since the subject of this research is how antiepileptic drugs affect skeletal growth in children, the discussion should be focused on the endochondral ossification of the growth plate rather than prenatal development.

R: Thank you for this suggestion. We have deleted the original text in lines 313-318, and replaced it with the following new sentences in lines 321-330 of the revised manuscript:

In basic terms, the mechanisms of bone formation involve both endochondral and intramembranous ossification. The former contributes to the growth of the axial and appendicular bones, while the latter is related to the growth of the craniofacial bones [48]. Endochondral ossification, which continues throughout the period of growth, entails sequential biological processes, in which chondrocytes proliferate and undergo hypertrophy to generate new cartilage at the growth plate of a long bone [49]. Simultaneously, the growth plate is invaded from the bony metaphysis by blood vessels and bone cell precursors, which remodel the cartilage into bone. Such remodeling appears to be triggered by apoptosis or cell death of the hypertrophic chondrocytes adjacent to the metaphyseal bone. Overall, the coordinated processes of chondrogenesis and ossification lead to long bone growth [50].

Q: Valproic acid is known to reduce chondrocyte hypertrophy (J Child Neurol. 2004;19(1):26-30), which conflicts with authors' suggestion. The prior studies should be comprehensively considered to discuss the differences between the results.

R:

In comparison with one model by Wu et al[1] in more detail which carried out the effects of VPA in prenatal stage with reduction of chondrocyte hypertrophy, our study was based on clinical data presented in our previous work[2], which showed that VPA could reduce growth in children. The current study goes further to examine the effects of VPA in childhood with reduction of chondrocyte proliferation. The hypertrophic mechanisms involved by VPA (in different stage of anatomic site) need to be further elucidated. A comparison table of these two papers is shown below.

Wu et al[1]

Fan et al

Purpose

To investigate the teratogenic effects of VPA on bone in the fetal stage

To investigate the effects of AEDs on the proliferation of the growth plate chondrocytes in childhood

Experimental materials

The second, third, and fourth metatarsal bones were isolated from Sprague-Dawley rat embryos at 20 days’ postconception

Chondrocytes were isolated from the epiphyseal growth plate of the tibia of male 3-week-old Sprague–Dawley rats

AEDs

VPA

VPA, OXA, LEV, LTG, TPM

VPA concentrations

0.1, 0.3, and 1 mmol/L

(liquid form)

30, 60, 150, 300, 450, 600 μg/mL

Target concentration of VPA

1mmol/L

60ug/ml (0.36mmol/L)

Treatment duration

7 days

4 to 5 days

Main findings

VPA markedly suppressed metatarsal longitudinal growth.  VPA-treated bones exhibited narrow growth plate proliferative and hypertrophic zones and an expanded ossification center

VPA markedly reduced the number of growth plate chondrocytes

Possible mechanisms

1. VPA significantly reduced total 3H-thymidine incorporation into the metatarsal rudiments

2. A reduced number of chondrocytes became hypertrophic owing to decreased proliferation, and an increased number of hypertrophic chondrocytes became apoptotic.

3. VPA–mediated increased apoptosis in the hypertrophic zone would lead to accelerated calcification of the growth plate

1. VPA up-regulated COX-2 mRNA and protein likely by increasing histone acetylation

2. VPA increased activation of caspase 3, leading to apoptosis

3. VPA increased apoptosis in a COX-2 dependent manner

Note :Epilim liquid (VPA; 200mg/ml=1.2M, which is much higher than the concentration used in Wu’s paper) (https://www.sanofi.co.uk/en/about-us/Healthcare-Solutions/products), is a liquid form of VPA.

According to the reviewer’s suggestion, we have added new sentences to explain the differences between two papers in lines 314-320.

Q: It is necessary to describe the limitation of the monolayer culture model (for example, a loss of polygonal morphology and dedifferentiation).

 R: We have added the limitations of this in vitro setting in the line 364-374.

This study had several limitations. First, rat growth plate chondrocytes cannot truly reflect in vivo human conditions. Second, the rat growth plate chondrocytes were cultured in monolayer and were not supplemented with ascorbic acid and growth factors, possibly leading to a loss of polygonal morphology and dedifferentiation [65]. A 3D system with such supplementations may be a more physiological setting that better preserves the spatial organization of the growth plate than the monolayer culture system used in this study. Third, as in vitro settings cannot simulate complex situations such as the compartmentation and microarchitecture of a joint, using a combination of in vitro and in vivo animal experiments should be a better strategy for investigating the impacts of VPA on statural growth. These limitations may have led to some bias in analyzing the effects of VPA on the growth of growth plate chondrocytes in the present study.

Q: Conclusion section

The contents of Lines 370–377 and lines 379–387 overlap.

 R: Thank you for pointing that out. We have deleted them and further modified the conclusions section.

Minor points

Q: Running title (line 4): Use more specific verb than “impact”.

R: We have replaced “impact” with “inhibit” in the revised running title.

Q: Lines 198–199 and 317–318: Once an abbreviation has been defined, don't repeat the definition again.

R: Done.

Q: Lines 172–174: The sentences should be deleted (editing error).

R: Thank you for pointing that out. These sentences have been deleted in the new version of the manuscript.

Q: Lines 167, 249, and 298: The terms “standard error” and “standard deviation” were confused.

R: Thank you for pointing out those errors. We have corrected them in the new version of the manuscript.

Q: Figure 6: Replace 3nd, 4nd, and 5nd with 3rd, 4th, and 5th, respectively.

R: Thank you for pointing out those errors. We have now corrected them

Reference

  1. Wu, S.; Legido, A.; De Luca, F. Effects of valproic acid on longitudinal bone growth. J Child Neurol 2004, 19, 26-30, doi:10.1177/088307380401900105011.
  2. Lee, H.S.; Wang, S.Y.; Salter, D.M.; Wang, C.C.; Chen, S.J.; Fan, H.C. The impact of the use of antiepileptic drugs on the growth of children. BMC Pediatr 2013, 13, 211, doi:10.1186/1471-2431-13-211.

Reviewer 2 Report

The authors investigated the effect of five AED's including VPA, OXA, LEV, LTG, and TPM on growth plate chondrocytes as well as VPA's effects on cartilage matrix gene expression; chondrocyte apoptosis, non-cleaved and cleaved caspase 3 expression; COX-2 expression and histone acetylation in growth plate chondrocyte. They found that VPA but not other AEDs decreased chondrocyte number in growth plate after five days of treatment in a dose-dependent fashion. VPA also stimulated chondrocyte apoptosis, non-cleaved and cleaved caspase 3 expression; COX-2 expression, histone acetylation, and that VPA induced apoptosis is possibly via COX-2 dependent inflammation pathways.

This is a well-conducted study. It is also an interesting paper in that it showed negative effect of VPA on bone growth in cellular level. It provided new information at cellular level for the well known VPA-related decreased stature growth. In that sense, this is an important study and helps to fill the gap of knowledge in this field.

Minor revision:

1- Line 84: Typo- please correct Lamicta as Lamictal.

Author Response

Reviewer B

Comments and Suggestions for Authors

Q: The authors investigated the effect of five AED's including VPA, OXA, LEV, LTG, and TPM on growth plate chondrocytes as well as VPA's effects on cartilage matrix gene expression; chondrocyte apoptosis, non-cleaved and cleaved caspase 3 expression; COX-2 expression and histone acetylation in growth plate chondrocyte. They found that VPA but not other AEDs decreased chondrocyte number in growth plate after five days of treatment in a dose-dependent fashion. VPA also stimulated chondrocyte apoptosis, non-cleaved and cleaved caspase 3 expression; COX-2 expression, histone acetylation, and that VPA induced apoptosis is possibly via COX-2 dependent inflammation pathways.

This is a well-conducted study. It is also an interesting paper in that it showed negative effect of VPA on bone growth in cellular level. It provided new information at cellular level for the well known VPA-related decreased stature growth. In that sense, this is an important study and helps to fill the gap of knowledge in this field.

 R: Thank you. We are very exciting and appreciated by your excellent comments on our work. It is very hard to meet someone knows the importance of this study. Thank you again.

Q: Minor revision:

  • Line 84: Typo- please correct Lamicta as Lamictal.

R: done. Thank you.

Reviewer 3 Report

This is an interesting study in which rat growth plate chondrocytes were cultured to investigate the effects of some AEDs, such as valproic acid (VPA), oxcarbazepine (OXA), levetiracetam (LEV), lamotrigine (LTG), and topiramate (TPM) on the skeletal growth. The findings indicate that  only VPA markedly reduces the number of chondrocytes by apoptosis.

The paper is clearly written, discussion and conclusions are rather adequate.

Minor spell check is required. For example, in the Abstract (line 22) the sentence "How AEDs impact skeletal growth are not clear" should be changed in: "How AEDs impact skeletal growth is not clear". In the Conclusions (line 387) "...inhibitors may be used to ameliorated VPA-induced cell death" should be changed in: "...inhibitors 386 may be used to ameliorate VPA-induced cell death".

In the Introduction the authors state that "Our previous work has demonstrated that VPA treatment significantly impairs statural growth in pediatric patients with epilepsy". Is this the only pathophysiological mechanism for poor statural growth in children taking VPA? Are there other hormonal factors for that? In which percentage of children chondrocyte growth is the main reason for growth failure? The Author should specify that in the Introduction or in the Discussion.

The Author propose (see Abstract, Discussion - lines 375-377) that 
"...a variety of pharmacological agents such as COX-2 inhibitors or, potentially, histone deacetylase inhibitors may be used to mitigate the adverse effects of VPA on chondrocyte growth". The findings of this study are based on an animal model. Chronic use of COX-2 inhibitors has adverse effects, and it is unrealistic and dangerous to recommend the use of these drugs for many years in children taking VPA who are a very great number! So, I would suggest to the Author to mitigate and limit this conclusion, even recommending further research on humans needed to confirm this experimental finding

Author Response

Reviewer C

Q: This is an interesting study in which rat growth plate chondrocytes were cultured to investigate the effects of some AEDs, such as valproic acid (VPA), oxcarbazepine (OXA), levetiracetam (LEV), lamotrigine (LTG), and topiramate (TPM) on the skeletal growth. The findings indicate that only VPA markedly reduces the number of chondrocytes by apoptosis.

The paper is clearly written, discussion and conclusions are rather adequate.

R: Thank you.

Q: Minor spell check is required. For example, in the Abstract (line 22) the sentence "How AEDs impact skeletal growth are not clear" should be changed in: "How AEDs impact skeletal growth is not clear". In the Conclusions (line 387) "...inhibitors may be used to ameliorated VPA-induced cell death" should be changed in: "...inhibitors 386 may be used to ameliorate VPA-induced cell death".

R: Thank you for pointing out those issues. As per your suggestion, we have reviewed the text and corrected any errors.

Q: In the Introduction the authors state that "Our previous work has demonstrated that VPA treatment significantly impairs statural growth in pediatric patients with epilepsy". Is this the only pathophysiological mechanism for poor statural growth in children taking VPA? Are there other hormonal factors for that? In which percentage of children chondrocyte growth is the main reason for growth failure? The Author should specify that in the Introduction or in the Discussion.

R: Thank you for these clarifying questions. No, VPA treatment is not the only mechanism causing poor statural growth in children with epilepsy. Rather, VPA exposure is just one of several factors affecting such children with poor statural growth. Therefore, we have revised the sentence in question to “Our previous work has demonstrated that VPA treatment is associated with the impairment of statural growth….” to make the claim of the sentence less strong.

We have also added the following text in lines 48-54 of the revised manuscript: “The diagnosis and treatment of epilepsy most commonly occur in childhood when bone growth is at its maximum. The processes involved in linear growth are very complicated and are regulated by several factors, such as endocrine mechanisms, paracrine mechanisms, proinflammatory cytokines, cartilage extracellular matrix regulation pathways, etc. The unwanted effects of AEDs on bone health are likely to be related to linear growth. Findings such as low serum calcium and 25-hydroxyvitamin D (25[OH]D) levels as well as decreased bone mineral density (BMD) have been frequently reported in epileptic children treated with AEDs.”

 Q: The Author propose (see Abstract, Discussion - lines 375-377) that 
"...a variety of pharmacological agents such as COX-2 inhibitors or, potentially, histone deacetylase inhibitors may be used to mitigate the adverse effects of VPA on chondrocyte growth". The findings of this study are based on an animal model. Chronic use of COX-2 inhibitors has adverse effects, and it is unrealistic and dangerous to recommend the use of these drugs for many years in children taking VPA who are a very great number! So, I would suggest to the Author to mitigate and limit this conclusion, even recommending further research on humans needed to confirm this experimental finding

R: We agree with your comment and suggestion. Upon further review, we feel that the conclusion in question was overly lacking in specifics. We therefore revise it and of the surrounding context accordingly. Thank you for your insightful comments.

Round 2

Reviewer 1 Report

The authors responded well and I have no further comments.